# Associations between β-blockers and psychiatric and behavioural outcomes: A population-based cohort study of 1.4 million individuals in Sweden

**Yasmina Molero**[1,2], **Sam Kaddoura**[3,4,5], **Ralf Kuja-Halkola**[2], **Henrik Larsson**[2,6], **Paul Lichtenstein**[2], **Brian M. D'Onofrio**[2,7], **Seena Fazel**[8,9] *

1 Department of Clinical Neuroscience, Centre for Psychiatry Research, Karolinska Institutet, Stockholm, Sweden, 2 Department of Medical Epidemiology and Biostatistics, Karolinska Institutet, Stockholm, Sweden, 3 School of Medicine, Imperial College, London, United Kingdom, 4 Chelsea and Westminster Hospital, London, United Kingdom, 5 Royal Brompton Hospital, London, United Kingdom, 6 School of Medical Sciences, Örebro University, Örebro, Sweden, 7 Department of Psychological and Brain Sciences, Indiana University, Bloomington, Indiana, United States of America, 8 Department of Psychiatry, University of Oxford, Oxford, United Kingdom, 9 Oxford Health NHS Foundation Trust, Oxford, United Kingdom

* seena.fazel@psych.ox.ac.uk

## Abstract

### Background

β-blockers are widely used for treating cardiac conditions and are suggested for the treatment of anxiety and aggression, although research is conflicting and limited by methodological problems. In addition, β-blockers have been associated with precipitating other psychiatric disorders and suicidal behaviour, but findings are mixed. We aimed to examine associations between β-blockers and psychiatric and behavioural outcomes in a large population-based cohort in Sweden.

### Methods and findings

We conducted a population-based longitudinal cohort study using Swedish nationwide high-quality healthcare, mortality, and crime registers. We included 1,400,766 individuals aged 15 years or older who had collected β-blocker prescriptions and followed them for 8 years between 2006 and 2013. We linked register data on dispensed β-blocker prescriptions with main outcomes, hospitalisations for psychiatric disorders (not including self-injurious behaviour or suicide attempts), suicidal behaviour (including deaths from suicide), and charges of violent crime. We applied within-individual Cox proportional hazards regression to compare periods on treatment with periods off treatment within each individual in order to reduce possible confounding by indication, as this model inherently adjusts for all stable confounders (e.g., genetics and health history). We also adjusted for age as a time-varying covariate. In further analyses, we adjusted by stated indications, prevalent users, cardiac severity, psychiatric and crime history, individual β-blockers, β-blocker selectivity and solubility, and use of other medications. In the cohort, 86.8% (*n* = 1,215,247) were 50 years and over, and

**Data Availability Statement:** Data may be obtained from a third party and are not publicly available. The Public Access to Information and Secrecy Act in Sweden prohibits us from making individual level data publicly available due to ethical concerns about identification. Researchers who are

interested in replicating our work can apply for individual level data from: Statistics Sweden (mikrodata@scb.se) for data from the Total Population Register and the Longitudinal Integrated database for Health Insurance and Labour Market Studies; The Swedish National Council for Crime Prevention (statistik@bra.se) for data from the Register of People Suspected of Offences; the Swedish Prison and Probation Service (hk.fou@kriminalvarden.se) for data from the Prison and Probation Register; The National Board of Health and Welfare (registerservice@socialstyrelsen.se) for data from The Patient Register, The Prescribed Drug Register, and the Cause of Death Register.

**Funding:** This study was supported by the Wellcome Trust (No 202836/Z/16/Z): https://wellcome.org/grant-funding (SF), the Swedish Research Council for Health Working Life and Welfare (2015-0028): https://forte.se/en/ (PL and HL), the American Foundation for Suicide Prevention (DIG-1-037-19): https://afsp.org/research-grant-information (BMD), and Karolinska Institutet Funds (2016fobi50581): https://staff.ki.se/ki-foundations-funds-list-of-grants (YM). The funders had no role in study design, data collection and analysis, decision to publish, or preparation of the manuscript.

**Competing interests:** I have read the journal's policy and the authors of this manuscript have the following competing interests: HL reports receiving grants from Shire Pharmaceuticals; personal fees from and serving as a speaker for Medice, Shire/Takeda Pharmaceuticals and Evolan Pharma AB; and sponsorship for a conference on attention-deficit/hyperactivity disorder from Shire/Takeda Pharmaceuticals and Evolan Pharma AB, all outside the submitted work. All other authors have declared that no competing interests exist.

**Abbreviations:** ACE, angiotensin-converting enzyme; ATC, Anatomical Therapeutic Chemical; CI, confidence interval; HR, hazard ratio; ICD-10, International Classification of Diseases, 10th revision; RCT, randomised controlled trial; SSRI, selective serotonin-reuptake inhibitor.

52.2% ($n = 731,322$) were women. During the study period, 6.9% ($n = 96,801$) of the β-blocker users were hospitalised for a psychiatric disorder, 0.7% ($n = 9,960$) presented with suicidal behaviour, and 0.7% ($n = 9,405$) were charged with a violent crime. There was heterogeneity in the direction of results; within-individual analyses showed that periods of β-blocker treatment were associated with reduced hazards of psychiatric hospitalisations (hazard ratio [HR]: 0.92, 95% confidence interval [CI]: 0.91 to 0.93, $p < 0.001$), charges of violent crime (HR: 0.87, 95% CI: 0.81 to 0.93, $p < 0.001$), and increased hazards of suicidal behaviour (HR: 1.08, 95% CI: 1.02 to 1.15, $p = 0.012$). After stratifying by diagnosis, reduced associations with psychiatric hospitalisations during β-blocker treatment were mainly driven by lower hospitalisation rates due to depressive (HR: 0.92, 95% CI: 0.89 to 0.96, $p < 0.001$) and psychotic disorders (HR: 0.89, 95% CI: 0.85 to 0.93, $p < 0.001$). Reduced associations with violent charges remained in most sensitivity analyses, while associations with psychiatric hospitalisations and suicidal behaviour were inconsistent. Limitations include that the within-individual model does not account for confounders that could change during treatment, unless measured and adjusted for in the model.

## Conclusions

In this population-wide study, we found no consistent links between β-blockers and psychiatric outcomes. However, β-blockers were associated with reductions in violence, which remained in sensitivity analyses. The use of β-blockers to manage aggression and violence could be investigated further.

## Author summary

### Why was this study done?

- β-blockers are primarily cardiac medications that are widely used for treating anxiety and are also suggested for the management of clinical depression and aggression, although research on efficacy is conflicting and limited by small samples and methodological problems.

- β-blockers have been linked to an increased risk of suicidal behaviour, but findings are inconclusive.

- More evidence using large samples and appropriate designs is needed on real-world effects on mental health and behavioural outcomes in people taking β-blockers.

### What did the researchers do and find?

- We examined a population-based cohort of 1,400,766 persons in Sweden who had been treated with β-blockers using a within-individual design; i.e., we compared individuals to themselves during medication and non-medication periods to account for background factors that may confound associations.

- Periods on β-blocker treatment were associated with an 8% lower risk of being hospitalised due to a psychiatric disorder, a 13% lower risk of being charged with a violent crime by the police, and an 8% increased risk of being treated for suicidal behaviour or suicide mortality.

- Reduced associations with violent charges were consistent across sensitivity analyses, while associations with suicidal behaviour and psychiatric hospitalisations varied by specific psychiatric diagnoses, past psychiatric problems, and cardiac severity.

**What do these findings mean?**

- The widespread use of β-blockers to manage anxiety is not supported in this real-world study that examined presentations of anxiety in secondary care.

- Studies using other designs (e.g., randomised controlled trials) are needed to better understand the role of β-blockers in the management of aggression and violence.

- If findings on violence are confirmed by studies that use other designs, β-blockers could be considered to manage aggression and hostility in individuals with psychiatric conditions.

## Introduction

Beta adrenergic-blocking agents, or β-blockers, act by blocking circulating neurotransmitter catecholamines norepinephrine (noradrenaline) and epinephrine (adrenaline) from binding to adrenoreceptors, thus reducing heart rate and blood pressure [1]. They are primarily used to treat hypertension, angina, heart failure, and arrhythmias, and for the secondary prevention of cardiovascular events. β-blockers also have other indications, including migraine, essential tremor, hyperthyroidism, and glaucoma [1].

Although β-blockers have no clear psychiatric indications, they are widely prescribed for treating anxiety [2]. However, there have been concerns of psychiatric adverse events during β-blocker use [3], and sleep disturbances, psychoses, and depression are listed as potential adverse events in the summary of product characteristics for β-blockers [4]. This is supported by observational studies that found an increased risk of depression for patients using β-blockers [5–10]. Several case reports have linked β-blockers to psychosis and delirium but there are no larger studies on these outcomes. Observational studies have also found an increased risk of suicide among individuals taking β-blockers as compared to controls [11–13]. However, there is contrasting evidence; β-blockers have been associated with decreases in depression and anxiety in other observational investigations [9,10,14–19] and a randomised controlled trial [20]. In addition, there are several observational studies [2,3,21–25] and randomised controlled trials [26–30] showing no associations with psychiatric events.

Inconsistencies across observational studies could be due to differences in case definition, varying measures of psychiatric outcomes, small and selected samples, short-term follow-up, and limited adjustment of confounding factors [3]. Importantly, observational studies have compared β-blocker users to non-users and are thus limited by confounding by indication (i.e., that the reason for prescribing the medication is also associated with the outcome).

Interpreting results from randomised controlled trials is also complicated as most trials have been short term, underpowered to detect rare but serious events, have not used standardised instruments to measure psychiatric outcomes, have excluded patients with psychiatric history, or had a high risk of bias [3,28,31].

Furthermore, β-blockers are classed into lipid solubility, hydrophilic and lipophilic (or hydrophobic), and by selectivity, where some are non-selective and others are selective for β1-adrenoceptors. It has been proposed that β-blockers may be differently associated with psychiatric and behavioural outcomes depending on their classification [32], yet only a limited range of β-blockers (mostly propranolol and pindolol) have been included in previous studies. Moreover, most studies have focused on depressive and anxiety disorders and have not examined associations with a wider range of psychiatric outcomes.

In addition to treating some psychiatric symptoms, β-blockers are also used in the clinical treatment of behavioural problems such as aggression and violence in individuals with certain psychiatric or neurological conditions [33], including schizophrenia [34,35], autism spectrum disorders [36], attention-deficit hyperactivity disorder [37], dementia [38], intellectual disability disorders [39,40], and traumatic brain injury [41,42]. Although expert opinion suggests that they are effective [33,42,43], evidence is of low quality and based almost entirely on small uncontrolled studies with short follow-up [39]. Their effectiveness in other patient groups (i.e., without psychiatric or neurological conditions) has not been examined.

Given the widespread use of β-blockers [44,45], well-designed studies that examine associations with psychiatric and behavioural outcomes—both in patients who are prescribed these medications to treat psychiatric and behavioural symptoms and in patients prescribed for cardiac or other indications—are necessary. This is particularly important because psychiatric problems are common in individuals with cardiac conditions; 1 in 5 patients with heart failure suffers from depression, with higher prevalence rates (up to 42%) in those with more severe heart failure [46] and around 30% report clinically significant levels of anxiety [47]. Patients with heart failure also have a nearly 2-fold increased risk of dying from suicide in the months following the heart failure [48]. A further increased risk of psychiatric and suicidal events during β-blocker treatment would raise concerns about medication safety. Then again, β-blockers may be underutilised [49] because evidence on safety is conflicting and largely limited by methodological weaknesses. Thus, more research including large samples and appropriate designs is needed to provide guidance on medication benefits and safety in treatment decisions.

We examined associations between β-blocker use and psychiatric and behavioural outcomes, including hospitalisations for psychiatric disorders, suicidal behaviour and deaths from suicide, and charges of violent crime, by applying a within-individual design (i.e., we compared individuals to themselves during medication and non-medication periods [50]) in a population-based cohort of 1.4 million β-blocker users who were followed for 8 years.

## Materials and methods

### Design

We conducted a population-based longitudinal cohort study using Swedish nationwide registers linked through each person's unique identification number [51]. Registers included the Total Population Register (for information on age, sex, and migration), the Swedish Prescribed Drug Register (for information on dispensed medications), the Swedish Patient Register (for information on diagnoses, hospitalisations, and treatment of suicidal behaviour), the Cause of Death Register (for information on death by suicide and other causes), the Register of Persons Suspected of Offences (for information on charges for violent and non-violent crime), the Longitudinal Integrated Database for Health Insurance and Labour Market Studies (LISA; for

information on civil status and source of income), and the Prison and Probation Services Register (for information on periods in prison) [51–57]. For more details on the registers, see S1 Text, page 2. We applied a within-individual design [58] that inherently adjusts for all stable confounders, i.e., factors that do not change during the study period (e.g., genetics and health history), and more fully adjusts for stable factors associated with confounding by indication.

## Participants and setting

We identified all individuals with dispensed β-blockers (i.e., filled-in prescriptions) in the Swedish population aged 15 and older (i.e., the age of criminal responsibility). Data on medication exposure in the Prescribed Drug Register was available from July 1, 2005; however, all information on each collected prescription was not complete in 2005 [59]. The study period therefore started in January 1, 2006 and ended in December 31, 2013 (the last available date for register linkage).

## Medications

β-blockers were defined as beta-adrenergic blocking agents (Anatomical Therapeutic Chemical [ATC] classification system: C07AA03, C07AA05, C07AA07, C07AB02, C07AB03, C07AB07, C07AG01, C07AG02) and included atenolol, bisoprolol, carvedilol, labetalol, metoprolol, pindolol, propranolol, and sotalol. Data on dispensed medications were extracted from the Swedish Prescribed Drug Register, which includes information on all prescriptions that are dispensed from all pharmacies in Sweden, and has less than 0.3% missing information [54]. All Swedish residents have subsidised medications via a common non-claim health care insurance. Treatment periods were defined as at least 2 consecutive dispenses within 6 months to ensure treatment continuity (as in previous studies) [60,61]. This span was chosen as the Swedish Pharmaceutical Benefits allows for a maximum of 3 months' supply for each prescription [62]. This meant that individuals who collected prescriptions within this span were considered to be under treatment; their treatment period started on the date of their first dispensed medication and ended on the date their last dispense within this span. Dispenses more than 6 months apart from the last dispense were considered to be the start of a new treatment period. Individuals who collected a single prescription may or may not have taken the medication. To address uncertainty over medication adherence, we excluded them from our primary analyses. However, this could potentially increase the risk of survival bias (i.e., that individuals who collected a single β-blocker prescription may have stopped taking the medication due to adverse events, while those who collected several β-blocker prescriptions had fewer adverse events and thus continued taking the medication) and direct associations towards the null. We therefore carried out sensitivity analyses where we included those who had collected a single prescription. Furthermore, we had excluded individuals with the instructions in the prescription text to take the medications "pro re nata" (PRN; i.e., as required) from our cohort due to uncertainty over regular medication use. However, this could increase the risk of selection bias, as a proportion of these individuals may have been prescribed β-blockers to treat anxiety. We therefore carried sensitivity analyses including them.

Initially, we identified 1,628,655 individuals who had been dispensed a β-blocker between 2006 and 2013. We excluded individuals with other treatment patterns (S1 Fig), such as individuals who collected a single prescription (*n* = 134,336); individuals PRN instructions (*n* = 64,822); individuals under age 15, i.e., under the age of criminal responsibility in Sweden (*n* = 2,729); and individuals with irregularly collected prescriptions, i.e., where new prescriptions were collected more than 6 months after the previous one (*n* = 26,002). The final cohort included 1,400,766 individuals.

## Psychiatric and behavioural outcomes

Outcomes included: (1) hospitalisations due to a psychiatric disorder (International Classification of Diseases, 10th revision [ICD-10]: F10-F69, F80-F99, excluding organic and intellectual disability disorders, and self-injurious behaviour or suicide attempts); (2) death from suicide or unplanned (i.e., without prior appointment or referral) hospital and specialised outpatient care visits due to self-injurious behaviour or suicide attempt as registered in mortality or patient records (ICD-10: X60-X84); and (3) charges of violent crime (i.e., crimes against people in the Swedish penal code) after a completed investigation by police, prosecution service, or customs authority. We used the incident date of the violent crime (i.e., the date when the crime was committed) rather than the date of the charge. Each event was treated as a distinct observation, meaning that individuals could experience repeated events of the same outcome. If more than 1 event of the outcome of interest was registered on the same day (e.g., more than 1 violent crime), only 1 event was counted that day. Data were collected from the National Patient Register (outcomes 1 to 2) [53], the Cause of Death Register (outcome 2) [52], and the Register of People Suspected of Offences (outcome 3) [56]. For more details on outcomes, see S1 Text, page 3.

## Secondary outcomes

We also examined 5 secondary outcomes to test the robustness of results. Secondary outcomes included: (1) hospitalisations due to psychotic disorders (ICD-10: F20-F29); (2) hospitalisations due to depressive disorders (ICD-10: F32-F34, F38-F39); (3) hospitalisations due to anxiety disorders (ICD-10: F40-F45, F48); (4) unplanned specialised outpatient care visits (as opposed to hospitalisations) due to a psychiatric disorder (ICD-10: F10-F69, F80-F99); and (5) charges of non-violent crime (i.e., all crimes other than violent crimes). Data were collected from the National Patient Register (outcomes 1 to 4) and the Register of People Suspected of Offences (outcome 5). For more details, see S1 Text, page 3.

## Demographic and health characteristics of the cohort

Information on sex and age was collected from the Total Population Register [51], civil status and source of income from the LISA Register [57], and diseases of the circulatory system from the National Patient Register (for more details, see S1 Text, page 5).

## Statistical analyses

All individuals in the cohort were followed from the start of the study period (January 1, 2006), or the date of immigration to Sweden, and were censored at death, emigration, or the end of study period (December 31, 2013), whichever occurred first. All time was split into periods of treatment and non-treatment. We removed periods where medication exposure and/or outcomes may not have been captured in the registers to account for time at risk, including periods in prison (extracted from the Prison and Probation Services Register).

Our null hypothesis was that no associations would be demonstrated between β-blockers and psychiatric hospitalisations, suicidal behaviour, and violent crime. A within-individual design—using stratified Cox proportional hazards regression—was applied to examine associations. The reason for using a within-individual design rather than standard between individual design, was that the between-individual design is liable to individual-specific unmeasured confounders that affect both the selection into β-blocker treatment and the tested outcomes. The current study design is a form of self-controlled case series design where each individual is entered as a separate stratum in the stratified Cox regression, and periods on medication are

compared to periods off medication within the same individual [50]. Mathematically, the model is given by

$$\lambda(t_{ij}|P_{ij}, X_{ij}, individual\ i) = \lambda_{0i}(t_{ij})e^{\beta P_{ij} + \gamma X_{ij}},$$

where $\lambda(t_{ij}|P_{ij}, X_{ij}, individual\ i)$ is the conditional hazard function at time $t_{ij}$, given $P_{ij}$, $X_{ij}$ (where $P_{ij}$ is the exposure and $X_{ij}$ the vector of measured covariates), and the individual $i$. By conditioning on the individual, and assuming individual-specific baseline hazards (the $\lambda_{0i}(t_{ij})$ in the equation), the model implicitly adjusts for all stable (i.e., time-invariant) confounders that are not readily observed in register data (such as genetic and other background risk factors) within the individual; these are absorbed by the individual-specific baseline hazard. This design also allowed us to adjust more fully for confounding by indication that was stable during the study period. In the analyses, the underlying time scale was divided into several periods; each individual was followed from the start of the period (time zero) until treatment switch (i.e., from no treatment to treatment or vice versa), the occurrence of an event (outcome), or they became 1 year older in age, whichever came first (consequently, each period could be up to 365 days). After this, a new period started, time was reset to zero, and the individual was followed up until treatment switch, event, or next birthday. This was done until the individuals were censored at death, emigration, or the end of study period. Each time-to-event was thus treated as a distinct observation. Because time-at-risk was measured from the start of all periods, recurrent events were accounted for.

The within-individual design does, however, not adjust for time-varying factors, i.e., those that changed during follow-up (e.g., age, health status, or use of other medications). We therefore also adjusted for age as a continuous time-varying covariate at the start of each time period in all our analyses. We also used a quadratic function of age as a time-varying covariate at the start of each time period to allow for nonlinear effects in all our analyses. The within-individual model has been applied in several studies of associations between medications and psychiatric and behavioural outcomes [60,63], and underlying methods are discussed elsewhere [58,64]. We did not test for proportional hazards as they were expected to vary over follow-up. To estimate cause-specific hazard ratios (HRs), we treated the competing event of death as a censoring event, rather than fitting competing risks models (see S1 Text, page 6, for more details).

In the within-individual design, all individuals in the cohort are included in the analyses. However, only individuals who are discordant on medication status (i.e., change from on treatment to off treatment or vice versa) and experience an event, contribute directly to the estimate of medication exposure on the outcome (in the β-blocker cohort, 1,373,901 individuals [98.1%] changed medication status at least once during the study period; see Table 1 for more details on exposure and outcomes). All other individuals contribute indirectly to this estimate through the association of other time-varying covariates that are adjusted for in the model, such as age (see S1 Text, page 6, for more information on this design).

First, we analysed the cohort as a whole. In further analyses, we stratified on the indication for the prescription, i.e., the reason stated by the prescribing physician in the prescription text, using data mining methods for unstructured text (see S1 Text, page 5 and S3 Table). We categorised indications into 3 categories adopting a hierarchical and mutually exclusive approach: (1) psychiatric or behavioural; (2) cardiac; and (3) other or unspecified indications. We also stratified on β1 selectivity (β1 selective and non-selective β-blockers), solubility (hydrophilic and lipophilic β-blockers), and on individual β-blockers (atenolol, bisoprolol, carvedilol, metoprolol, propranolol, and sotalol—treatment periods with labetalol and pindolol included did not include enough outcome events to allow for separate analyses). See S1 Text, page 2.

**Table 1. Baseline characteristics of the study cohort.**

| **Demographic characteristics at the start of the study period (2006) (_n_ = 1,400,766)** | |
| --- | --- |
| _Sex_ | |
| Females | 52.2% (731,322) |
| Males | 47.8% (669,444) |
| _Age_ | |
| Under 30 | 1.7% (24,011) |
| 30–49 | 11.5% (161,508) |
| 50–69 | 44.4% (622,083) |
| 70 and older | 42.4% (593,164) |
| _Civil status[†]_ | |
| Married | 51.4% (720,223) |
| Divorced or widowed | 31.3% (438,400) |
| Unmarried | 14.4% (201,840) |
| _Source of income[†]_ | |
| Employed | 33.3% (466,025) |
| Educational grant | 0.9% (12,863) |
| Pension | 57.6% (806,332) |
| Disability pension | 11.7% (164,370) |
| Unemployment benefits | 3.5% (49,544) |
| Receiving state benefits | 2.6% (35,972) |
| **Treatment characteristics during the study period (2006–2013) (_n_ = 1,400,766)** | |
| _Individual β-blockers[††]_ | |
| Atenolol | 24.8% (347,540) |
| Bisoprolol | 16.6% (232,688) |
| Carvedilol | 2.1% (29,184) |
| Labetalol | 0.4% (6,182) |
| Metoprolol | 60.0% (826,165) |
| Pindolol | 0.7% (9,932) |
| Propranolol | 6.5% (91,214) |
| Sotalol | 2.9% (40,630) |
| _β1 selectivity[††]_ | |
| β1 selective β-blockers | 91.8% (1,285,297) |
| Non-selective β-blockers | 12.5% (175,281) |
| _Solubility[††]_ | |
| Hydrophilic β-blockers | 27.5% (385,711) |
| Lipophilic β-blockers | 79.4% (1,111,605) |
| _Indication for the prescription_ | |
| Psychiatric or behavioural indications | 1.1% (16,018) |
| Cardiac indications | 84.8% (1,187,137) |
| Other or unspecified indications | 14.6% (197,611) |
| **Diseases of the circulatory system during the study period (2006–2013)[†††] (_n_ = 1,400,766)** | |
| Acute rheumatic fever | 0.0% (251) |
| Chronic rheumatic heart diseases | 0.5% (6,502) |
| Hypertensive diseases | 49.9% (699,147) |
| Ischaemic heart diseases | 28.1% (393,194) |
| Pulmonary heart disease and diseases of pulmonary circulation | 2.6% (36,774) |
| Other forms of heart disease | 37.5% (525,586) |
| Cerebrovascular diseases | 12.6% (176,778) |

(_Continued_)

**Table 1.** (Continued)

| | |
|---|---|
| Diseases of arteries, arterioles, and capillaries | 7.3% (102,568) |
| Diseases of veins, lymphatic vessels, and lymph nodes | 7.8% (109,084) |
| Other and unspecified disorders of the circulatory system | 2.7% (38,443) |
| **Outcomes during the study period (2006–2013) (*n* = 1,400,766)** | |
| *Main outcomes* | |
| Any psychiatric hospitalisation | 6.9% (96,801) |
| Any suicidal behaviour | 0.7% (9,960) |
| Any violent crime | 0.7% (9,405) |
| *Secondary outcomes* | |
| Hospitalisations for psychotic disorders | 0.6% (8,459) |
| Hospitalisations for depressive disorders | 2.4% (33,364) |
| Hospitalisations for anxiety disorders | 1.9% (26,121) |
| Outpatient psychiatric visits (emergency visits only)†††† | 3.9% (54,562) |
| Non-violent crime | 3.8% (53,394) |
| *Number of events (off/on β-blockers)* | |
| Psychiatric hospitalisations | 245,457 (131,731/113,726) |
| Suicidal behaviour | 17,709 (11,034/6,675) |
| Violent crime | 16,825 (11,203/5,622) |
| *Individuals with outcome event and treatment status change†††††* | |
| Psychiatric hospitalisations | 6.6% (89,933) |
| Suicidal behaviour | 0.7% (9,552) |
| Violent crime | 0.7% (9,235) |
| **Use of other medications during the study period (2006–2013) (*n* = 1,400,766)** | |
| Antidepressants | 31.3% (438,548) |
| Antipsychotics | 7.3% (101,736) |
| Benzodiazepines | 43.9% (615,159) |
| Calcium channel blockers | 42.9% (600,663) |
| Renin-angiotensin system acting agents | 63.3% (886,580) |
| Statins | 49.8% (697,795) |
| Polypharmacy | 56.4% (790,237) |

[†] Missing information for 40,303 individuals. Note: income categories are not mutually exclusive.

[††] Not mutually exclusive categories: 182,769 individuals (13.1% of the cohort) were treated with 2 or more β-blocker during the study period, 59,812 (4.3%) were treated with both β1 selective and non-selective β-blockers, and 96,550 (6.9%) were treated with both hydrophilic and lipophilic β-blockers.

[†††] Not mutually exclusive categories.

[††††] Includes unplanned visits only, i.e., no referrals or previously appointed visits.

[†††††] Individuals with at least 1 event of the outcome in question who also changed medication status at least once during the study period (from on treatment to off treatment or vice versa).

## Sensitivity analyses—Alternative exposures and outcomes

We carried out several data-driven sensitivity analyses with alternative exposures and secondary outcomes to test the robustness of the results. To address the possibility of time-varying confounding effects due to an increased risk of psychiatric outcomes in the initial phase following a cardiac event [65,66], we excluded the first 3 months of the incident β-blocker treatment. Because psychiatric disorders also increase the risk of experiencing a cardiac event [67], we subsequently excluded the 3 months leading up to the incident β-blocker treatment. In our main analyses, we defined the end of a treatment period as the day of the last dispensed

prescription. This gives a more conservative estimate of medication exposure and does not account for late treatment or discontinuation effects. We therefore carried out sensitivity analyses where we extended each medication period by adding 3 months after the last dispensed prescription within that period. We then used antihistamines for systemic use (see S1 Text, page 4) as an independent exposure in the β-blocker cohort to examine nonspecific treatment effects, such as increased contacts with healthcare during medication periods. We also examined 5 secondary outcomes, including hospitalisations due to psychotic disorders, hospitalisations due to depressive disorders, hospitalisations due to anxiety disorders, outpatient care visits due to a psychiatric disorder, and charges of non-violent crime.

## Sensitivity analyses—Alternative samples

We carried out further data-driven sensitivity analyses with alternative samples. To address prevalent user bias (i.e., that a proportion of individuals in the cohort were already using β-blockers at the start of the study period and were therefore not liable to effects in the early phase of treatment), we excluded prevalent users, i.e., we examined only those who initiated treatment from January 1, 2007 onwards. Because β-blockers combined with selective serotonin-reuptake inhibitors (SSRIs) have been linked to reduced depression [68,69], we addressed the confounding effects of antidepressant use in sensitivity analyses. In these analyses, we excluded all individuals who had collected an antidepressant (i.e., an SSRI or another antidepressant, ATC: N06A) during the study period (i.e., 2006 to 2013) from the cohort and examined associations between β-blockers and outcomes in those who remained (i.e., those who had not collected an antidepressant during the study period). We also carried out analyses where we excluded individuals who had collected an antipsychotic medication (ATC: N05A), or common hypertension medications including calcium channel blockers (ATC: C08), renin-angiotensin system acting agents (ATC: C09), or statins (ATC: C10AA), respectively, to address confounding effects by other medications on psychiatric and behavioural outcomes. We addressed the issue that individuals with severe cardiac conditions could be more likely to experience a psychiatric outcome by stratifying analyses on individuals who had been hospitalised for cardiac conditions (ICD-10: I00-I99) within 1 year of the start of the first medication period and all other individuals (including both those who had received outpatient treatment for cardiac conditions and those not diagnosed with cardiac conditions within 1 year of the first medication period). We also accounted for previous psychiatric problems by stratifying analyses on individuals with a history of psychiatric problems (i.e., those who had been treated for psychiatric disorders and/or suicidal behaviour before the start of the study period) and all other individuals. We further examined associations with violent outcomes by including only individuals with a history of violent crime (i.e., before the start of the study period).

## Sensitivity analyses—Post hoc analyses

We carried out several post hoc sensitivity analyses to further test the robustness of results. We examined nonspecific treatment effects by using a different negative control medication—angiotensin-converting enzyme (ACE) inhibitors (ATC: C09AA)—as an independent exposure in the β-blocker cohort (see S1 Text, page 4 for details). Furthermore, we carried out analyses where we excluded individuals who had been prescribed benzodiazepines (ATC: N03AE, N05BA, N05CD, N05CF) to address the confounding effects of concurrent benzodiazepine use on psychiatric and behavioural outcomes. We also controlled for the confounding effects of polypharmacy by excluding individuals who had been prescribed 5 or more different medication classes during the same calendar year (see S1 Text, page 3). In our main analyses, we excluded individuals who collected single β-blocker prescriptions during follow-up

($n$ = 134,336). To address the possibility that these individuals may have stopped taking the medication due to adverse events, we carried out analyses including them. In these analyses, individuals with a single prescription were assumed to be exposed to medication during the 3 months following their collected prescription. In the main analyses, we also excluded individuals who had been instructed to take the medication as required (PRN) in the prescription text due uncertainty of regular β-blocker use. Because a proportion of these individuals may have been prescribed β-blockers to treat anxiety, we also carried out analyses including them in our main cohort. In these analyses, medication exposure for individuals with PRN instructions was modelled as in our main models (see Medications paragraph). To examine if β-blockers were differentially associated with violent crimes by age, we stratified individuals into different age groups depending on their age during the study period, up to age 30, age 30 to 49, age 50 to 60, and age 70 and older. We then examined associations between β-blockers and violent crime separately for each age group.

HRs and 95% confidence intervals (CIs) are presented for all analyses. We used SAS version 9.4 for all analyses. This study is reported as per the Strengthening the Reporting of Observational Studies in Epidemiology (STROBE) guideline (S1 Checklist).

### Ethical approval

This study was approved by the Swedish Ethical Review Authority (2013/5:8) in written form. The Swedish Ethical Review Authority waived the need for informed consent due to the register-based design. The study follows the Declaration of Helsinki.

## Results

### Selection of cohort

We identified 1,628,655 individuals who had been prescribed β-blockers during the study period between 2006 and 2013. After exclusions due to irregular medication use and age (S1 Fig), the final cohort included 1,400,766 individuals (15.7% of the total population of Sweden aged 15 years or older during the study period [$n$ = 8,945,456]).

### Characteristics of the β-blocker cohort

In the cohort, 52.2% ($n$ = 731,322) were women (Table 1). At the start of the study period, 1.7% ($n$ = 24,011) of the cohort were under age 30, 11.5% ($n$ = 161,508) were between age 30 and age 49, 44.4% ($n$ = 622,083) were between age 50 and age 69, and 42.4% ($n$ = 593,164) were age 70 and older. The most commonly diagnosed cardiac conditions during the study period included hypertensive diseases (49.9%, $n$ = 699,147), ischaemic heart diseases (28.1%, $n$ = 393,194), and other heart diseases (37.5%, $n$ = 525,586). The most commonly prescribed β-blocker was metoprolol, prescribed to 60.0% ($n$ = 826,165), followed by atenolol (24.8%, $n$ = 347,540) and bisoprolol (16.6%, $n$ = 232,688); 13.1% ($n$ = 182,769) of the cohort were treated with 2 or more different β-blockers. The large majority of prescriptions were for β1 selective β-blockers (91.8%, $n$ = 1,285,297). In terms of solubility, most prescriptions (79.4%, $n$ = 1,111,605) were for lipophilic β-blockers. We also examined the stated indications for the prescription and found that the majority of the cohort (84.8%; $n$ = 1,187,137) were prescribed β-blockers for a cardiac indication, 1.1% ($n$ = 16,018) for a psychiatric or behavioural indication, and 14.6% ($n$ = 197,611) for another or unspecified indication. During the study period, 6.9% ($n$ = 96,801) of the β-blocker users were hospitalised for a psychiatric disorder, 0.7% ($n$ = 9,960) presented with suicidal behaviour (i.e., treatment at hospital or specialised outpatient care for self-injurious acts or suicide attempts, or deaths from suicide as the stated cause

of death), and 0.7% (*n* = 9,405) were charged with a violent crime (i.e., attempted, completed, and aggravated forms of murder, manslaughter, unlawful threats, harassment, robbery, arson, assault, assault on an official, kidnapping, stalking, coercion, and sexual offences) after a completed investigation by police, prosecution service, or customs authority. More data on treatment characteristics, outcomes, and use of other medications are presented in Tables 1 and in S1.

## Associations between β-blockers and psychiatric and behavioural outcomes

We carried out analyses comparing all treatment periods to all non-treatment periods within each individual using stratified Cox proportional hazards regression (Fig 1; event rates in Table 1). Results from our within-individual analyses showed that periods on β-blocker treatment were associated with a lower HR of psychiatric hospitalisations (HR = 0.92, 95% CI = 0.91 to 0.93, *p* < 0.001). We found increased hazards of suicidal behaviour during β-blocker treatment periods (HR: 1.08, 95% CI: 1.02 to 1.15, *p* = 0.013) and reduced hazards of violent crime (HR: 0.87, 95% CI: 0.81 to 0.93, *p* < 0.001). Unadjusted results for all within-individual analyses are presented in S4 Table.

## Associations between β-blockers and psychiatric and behavioural outcomes by the indication for the prescription

To further adjust for confounding by indication, i.e., that characteristics that lead an individual to be prescribed β-blockers may also predispose them for the outcome, we stratified our within-individual analyses by the indication for the prescription (Fig 2; event rates in S2 Table). In these analyses, there were no statistically significant associations with outcomes among individuals with psychiatric or behavioural indications. We found that individuals with cardiac indications followed similar patterns as the overall results, i.e., decreased hazards of psychiatric hospitalisations (HR: 0.91, 95% CI: 0.90 to 0.93, *p* < 0.001) and violent crime (HR: 0.85, 95% CI: 0.79 to 0.91, *p* < 0.001), and increased hazards of suicidal behaviour (HR: 1.10, 95% CI: 1.02 to 1.19, *p* = 0.012). Individuals with other or unspecified indications showed decreased hazards of psychiatric hospitalisations during β-blocker treatment periods (HR: 0.95, 95% CI: 0.92 to 0.99, *p* = 0.007) and no statistically significant associations with other outcomes.

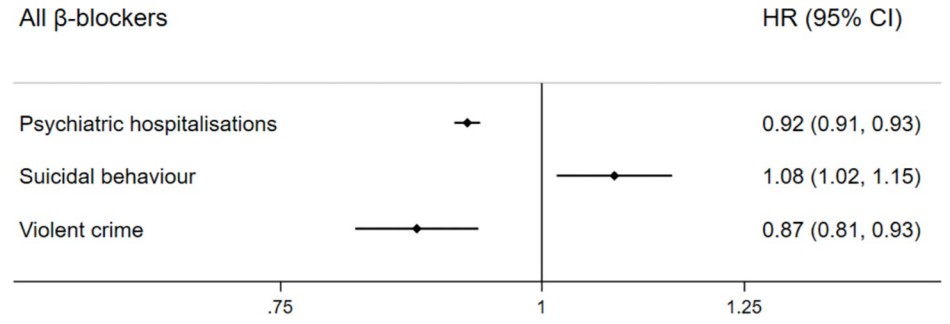

HR = Hazard ratio (represented by the dot); CI = Confidence interval (represented by the line)

**Fig 1. Age-adjusted within-individual associations between β-blockers and psychiatric and behavioural outcomes in the β-blockers cohort (*n* = 1,400,766).** CI, confidence interval; HR, hazard ratio.

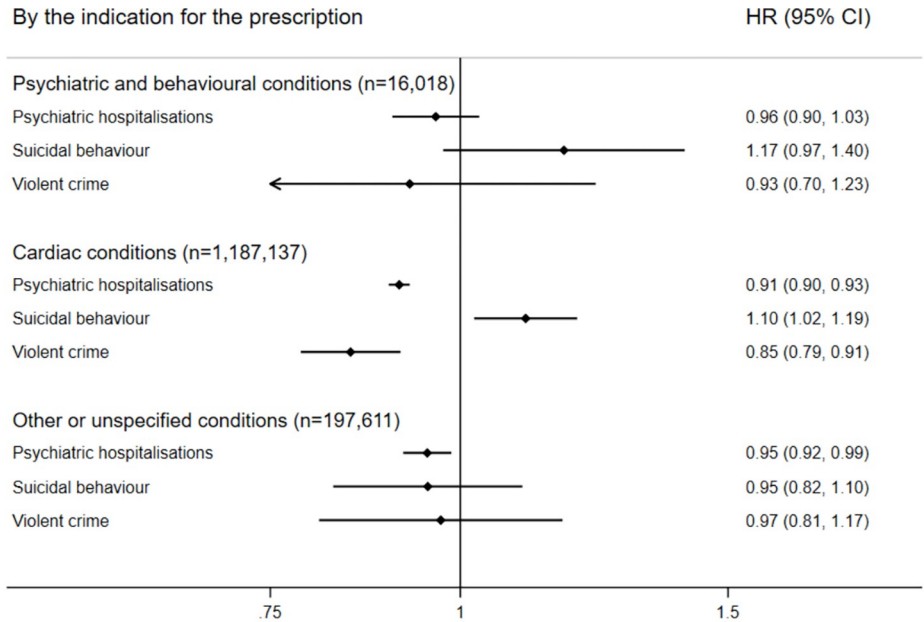

HR = Hazard ratio (represented by the dot); CI = Confidence interval (represented by the line)

**Fig 2. Age-adjusted within-individual associations between β-blockers and psychiatric and behavioural outcomes stratified by the indication for prescription.** CI, confidence interval; HR, hazard ratio.

## Associations between β-blockers and psychiatric and behavioural outcomes by β-1 selectivity and solubility, and by individual β-blockers

We stratified analyses by β1 selectivity (Fig 3; event rates in S2 Table). β1 selective β-blocker treatment periods were associated with reduced hazards of psychiatric hospitalisations and violent crime and were not associated with suicidal behaviour. Treatment periods with non-selective β-blockers were associated with reduced hazards of psychiatric hospitalisations but demonstrated no clear associations with the other outcomes. When stratified by solubility, treatment periods with hydrophilic β-blockers were associated with reduced hazards of psychiatric hospitalisations and violent crime, and no associations were shown for suicidal behaviour. Treatment periods with lipophilic β-blockers followed the same patterns as in the main analyses, i.e., reduced hazards of psychiatric hospitalisations and violent crime, and increased hazards of suicidal behaviour.

We also stratified analyses on individual β-blockers (S2 Fig; event rates in S2 Table). There were some variations between individual β-blockers; treatment with atenolol and metoprolol was associated with lower hazards of both psychiatric hospitalisations, and violent crime, and treatment with bisoprolol, propranolol, and sotalol were associated with reduced hazards of psychiatric hospitalisations.

## Sensitivity analyses—Alternative exposures and outcomes

We addressed the potential for time-varying confounding due to an increased risk of psychiatric sequelae following a cardiac event, by excluding the first 3 months of the incident β-blocker treatment period. Results remained similar to the main analyses (Table 2); decreased hazards of psychiatric hospitalisations and violent crime and increased hazards of suicidal behaviour

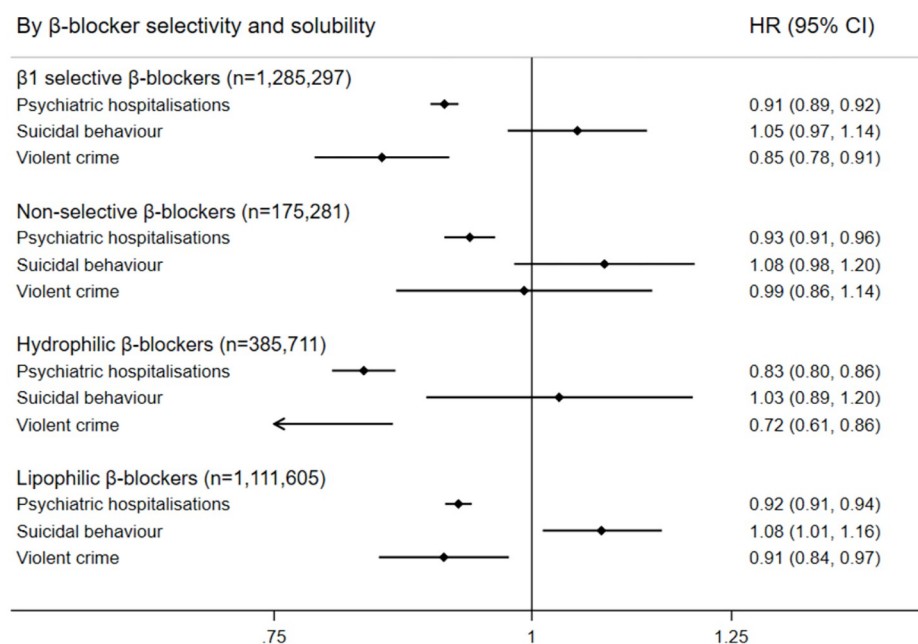

HR = Hazard ratio (represented by the dot); CI = Confidence interval (represented by the line)

**Fig 3. Age-adjusted within-individual associations between β-blockers and psychiatric and behavioural outcomes stratified by β-blocker selectivity and solubility.** CI, confidence interval; HR, hazard ratio.

during treatment. Because psychiatric disorders are associated with an increased risk of a subsequent cardiac event, we then excluded the 3 months leading up to the incident β-blocker treatment period to adjust for the effect of recent psychiatric or behavioural adverse events. Results from these analyses showed increased hazards of psychiatric hospitalisations (HR: 1.03, 95% CI: 1.02 to 1.05, $p < 0.001$) and suicidal outcomes (HR: 1.16, 95% CI: 1.09 to 1.24, $p < 0.001$), and reduced hazards of violent crime (HR: 0.89, 95% CI: 0.83 to 0.96, $p = 0.001$) during treatment. We then accounted for late treatment effects (e.g., discontinuation effects) by extending the end of a treatment period to 3 months after the last collected prescription. Results remained similar to the main analyses.

We also repeated our main models using 2 negative controls—antihistamines and ACE inhibitors—as independent exposures in the β-blockers cohort to examine nonspecific treatment effects. Results showed no associations with psychiatric hospitalisations (HR = 1.00, 95% CI = 0.95 to 1.05), suicidal behaviour (HR = 1.16, 95% CI = 0.99 to 1.36) or violent crime (HR = 1.23, 95% CI = 0.93 to 1.63) during antihistamine treatment periods, and increased hazards of suicidal behaviour (HR: 1.15, 95% CI: 1.02 to 1.31, $p = 0.020$), and no associations with psychiatric hospitalisations (HR: 0.99, 95% CI: 0.96 to 1.01, $p = 0.208$) or violent crime (HR: 1.01, 95% CI: 0.91 to 1.14, $p = 0.813$) during ACE inhibitor treatment periods.

We also carried out sensitivity analyses using secondary outcomes: hospitalisations for psychotic, depressive, and anxiety disorders, respectively. We found reduced hazards for hospitalisations for psychotic and depressive disorders during β-blocker treatment periods and no associations with anxiety disorder hospitalisations (Table 2). We further tested the robustness of results on psychiatric events by examining associations between β-blocker treatment and all outpatient visits, and there were no clear associations (HR: 0.99, 95% CI: 0.98 to 1.02, $p = 0.812$). We also examined treatment associations with non-violent criminal charges and

**Table 2. Sensitivity analyses; age-adjusted within-individual associations between β-blockers and psychiatric and behavioural outcomes using alternative exposures and outcomes.**

| | HR (95% CI) | Number of events | *P*-value |
|---|---|---|---|
| **Alternative exposures** | | | |
| *Excluding the first 3 months of the incident medication period (n = 1,400,766)* | | | |
| Psychiatric hospitalisations | 0.90 (0.89–0.91) | 103,850 | <0.001 |
| Suicidal behaviour | 1.08 (1.01–1.16) | 15,391 | 0.226 |
| Violent crime | 0.92 (0.85–0.98) | 14,916 | 0.017 |
| *Excluding the 3 months leading up to the incident medication period (n = 1,400,766)* | | | |
| Psychiatric hospitalisations | 1.03 (1.02–1.05) | 113,629 | <0.001 |
| Suicidal behaviour | 1.16 (1.09–1.24) | 17,129 | <0.001 |
| Violent crime | 0.89 (0.83–0.96) | 16,390 | 0.001 |
| *Adding 3 months after the last collected prescription (n = 1,400,766)* | | | |
| Psychiatric hospitalisations | 0.92 (0.91–0.94) | 245,457 | <0.001 |
| Suicidal behaviour | 1.08 (1.02–1.15) | 17,709 | 0.013 |
| Violent crime | 0.87 (0.81–0.93) | 16,825 | <0.001 |
| *Antihistamines as exposure (n = 117,373)* | | | |
| Psychiatric hospitalisations | 1.00 (0.95–1.05) | 32,546 | 0.898 |
| Suicidal behaviour | 1.16 (0.99–1.36) | 2,976 | 0.065 |
| Violent crime | 1.23 (0.93–1.63) | 1,795 | 0.154 |
| *ACE inhibitors as exposure (n = 561,868)* | | | |
| Psychiatric hospitalisations | 0.99 (0.96–1.01) | 94,805 | 0.208 |
| Suicidal behaviour | 1.15 (1.02–1.31) | 4,802 | 0.020 |
| Violent crime | 1.01 (0.91–1.14) | 6,293 | 0.813 |
| **Secondary outcomes (*n* = 1,400,766)** | | | |
| Hospitalisations for psychotic disorders | 0.89 (0.85–0.93) | 23,781 | <0.001 |
| Hospitalisations for depressive disorders | 0.92 (0.89–0.96) | 48,963 | <0.001 |
| Hospitalisations for anxiety disorders | 1.01 (0.97–1.05) | 45,321 | 0.552 |
| Outpatient treatment for psychiatric disorders | 0.99 (0.98–1.02) | 145,482 | 0.812 |
| Non-violent crime | 0.93 (0.91–0.95) | 122,974 | <0.001 |

ACE, angiotensin-converting enzyme; CI, confidence interval; HR, hazard ratio.

found that β-blocker treatment was associated with reduced hazards (HR: 0.93, 95% CI: 0.91 to 0.95, $p < 0.001$).

## Sensitivity analyses—alternative samples

We examined the risk of psychiatric and behavioural outcomes among new β-blocker users, i.e., those who had not used the medication before 2007 (Table 3). There was little difference with the overall findings (psychiatric hospitalisations: HR: 0.94, 95% CI: 0.92 to 0.96, $p < 0.001$; violent crime: HR: 0.88, 95% CI: 0.80 to 0.96, $p = 0.004$), although associations with suicidal behaviour did not reach statistical significance (HR: 1.09, 95% CI: 0.99 to 1.18, $p = 0.057$). To address a potential for survivor bias in our β-blocker cohort (i.e., that individuals who experienced adverse events discontinued with β-blockers), we carried out analyses where we included individuals who had collected only 1 prescription in the main β-blocker cohort. Results remained similar to the main analyses. In our main analyses, we had excluded individuals who had been instructed the medication PRN due to uncertainty of daily use. We carried out sensitivity analyses including them in the main β-blocker cohort, and results were similar (Table 3). To account for potentially confounding effects by other medications

**Table 3. Sensitivity analyses; age-adjusted within-individual associations between β-blockers and psychiatric and behavioural outcomes using alternative samples.**

| | HR (95% CI) | Number of events | |
|---|---|---|---|
| **Alternative samples** | | | |
| *Excluding prevalent users (n = 550,944)†* | | | |
| Psychiatric hospitalisations | 0.94 (0.92–0.96) | 131,239 | <0.001 |
| Suicidal behaviour | 1.09 (0.99–1.18) | 10,377 | 0.057 |
| Violent crime | 0.88 (0.80–0.96) | 11,332 | 0.004 |
| *Including those with only 1 dispense (n = 1,535,102)* | | | |
| Psychiatric hospitalisations | 0.94 (0.92–0.95) | 275,699 | <0.001 |
| Suicidal behaviour | 1.10 (1.04–1.17) | 21,113 | 0.001 |
| Violent crime | 0.88 (0.82–0.93) | 21,770 | <0.001 |
| *Including those with PRN\* instructions (n = 1,465,588)* | | | |
| Psychiatric hospitalisations | 0.92 (0.91–0.94) | 251,026 | <0.001 |
| Suicidal behaviour | 1.08 (1.02–1.15) | 18,621 | 0.012 |
| Violent crime | 0.87 (0.81–0.93) | 17,386 | <0.001 |
| *Excluding individuals with antidepressants (n = 962,218)* | | | |
| Psychiatric hospitalisations | 0.85 (0.83–0.88) | 58,054 | <0.001 |
| Suicidal behaviour | 1.07 (0.84–1.37) | 2,065 | 0.576 |
| Violent crime | 0.91 (0.82–1.00) | 7,307 | 0.059 |
| *Excluding individuals with antipsychotics (n = 1,299,030)* | | | |
| Psychiatric hospitalisations | 0.91 (0.89–0.93) | 127,216 | <0.001 |
| Suicidal behaviour | 1.08 (0.96–1.22) | 7,328 | 0.200 |
| Violent crime | 0.86 (0.80–0.93) | 12,010 | <0.001 |
| *Excluding individuals with benzodiazepines (n = 785,607)* | | | |
| Psychiatric hospitalisations | 0.84 (0.81–0.87) | 42,562 | <0.001 |
| Suicidal behaviour | 0.85 (0.65–1.10) | 1,905 | 0.216 |
| Violent crime | 0.91 (0.82–1.02) | 6,440 | 0.112 |
| *Excluding individuals with calcium channel blockers (n = 800,103)* | | | |
| Psychiatric hospitalisations | 0.91 (0.90–0.93) | 155,544 | <0.001 |
| Suicidal behaviour | 1.05 (0.98–1.13) | 12,627 | 0.162 |
| Violent crime | 0.88 (0.81–0.96) | 11,751 | 0.002 |
| *Excluding individuals with renin-angiotensin system acting agents (n = 514,186)* | | | |
| Psychiatric hospitalisations | 0.92 (0.90–0.94) | 110,848 | <0.001 |
| Suicidal behaviour | 1.07 (0.98–1.15) | 10,531 | 0.116 |
| Violent crime | 0.85 (0.77–0.93) | 8,265 | <0.001 |
| *Excluding individuals with statins (n = 702,971)* | | | |
| Psychiatric hospitalisations | 0.94 (0.92–0.95) | 140,800 | <0.001 |
| Suicidal behaviour | 1.08 (1.01–1.16) | 11,705 | 0.049 |
| Violent crime | 0.86 (0.79–0.94) | 10,973 | <0.001 |
| *Excluding individuals with polypharmacy (n = 610,529)* | | | |
| Psychiatric hospitalisations | 0.84 (0.82–0.87) | 61,151 | <0.001 |
| Suicidal behaviour | 1.04 (0.89–1.21) | 4,101 | 0.619 |
| Violent crime | 0.83 (0.75–0.92) | 7,591 | 0.001 |
| *By cardiac severity* | | | |
| *Including only individuals hospitalised for cardiac disorders†† (n = 278,429)* | | | |
| Psychiatric hospitalisations | 1.14 (1.12–1.17) | 85,297 | <0.001 |
| Suicidal behaviour | 1.20 (1.05–1.36) | 4,219 | 0.006 |
| Violent crime | 0.85 (0.73–0.98) | 3,570 | 0.022 |

*(Continued)*

**Table 3.** (Continued)

| | HR (95% CI) | Number of events | |
|---|---|---|---|
| *Excluding individuals hospitalised for cardiac disorders†† (n = 1,122,337)* | | | |
| Psychiatric hospitalisations | 0.80 (0.79–0.82) | 160,160 | <0.001 |
| Suicidal behaviour | 1.05 (0.97–1.13) | 13,490 | 0.222 |
| Violent crime | 0.88 (0.81–0.95) | 13,255 | 0.001 |
| **By previous history†††** | | | |
| *Including only individuals with a history of psychiatric disorders or suicidal behaviour (n = 92,619)* | | | |
| Psychiatric hospitalisations | 0.93 (0.92–0.95) | 137,081 | <0.001 |
| Suicidal behaviour | 1.11 (1.02–1.20) | 10,028 | 0.010 |
| Violent crime | 0.83 (0.76–0.91) | 7,502 | <0.001 |
| *Excluding individuals with a history of psychiatric disorders or suicidal behaviour (n = 1,308,147)* | | | |
| Psychiatric hospitalisations | 0.90 (0.88–0.92) | 108,376 | <0.001 |
| Suicidal behaviour | 1.03 (0.93–1.15) | 7,681 | 0.566 |
| Violent crime | 0.92 (0.83–1.01) | 9,323 | 0.078 |
| *Including only individuals with a history of violent crime (n = 6,902)* | | | |
| Violent crime | 0.82 (0.74–0.91) | 5,803 | <0.001 |
| *Violent crime by age categories* | | | |
| Under 30 (*n* = 24,011) | 0.78 (0.59–1.03) | 1,688 | 0.077 |
| 30–49 (*n* = 161,083) | 0.76 (0.56–1.03) | 7,318 | 0.073 |
| 50–69 (*n* = 622,083) | 0.79 (0.72–0.88) | 7,114 | <0.001 |
| 70 and older (*n* = 593,164) | 0.86 (0.80–0.92) | 705 | <0.001 |

[†] Including only individuals who initiated β-blocker treatment from January 1, 2007 and onwards.

[††] During the first year after medication initiation.

[†††] Before the start of the study period, i.e., January 1, 2006.

[*] PRN = Pro re nata, i.e., instructed to take medications "as required."

CI, confidence interval; HR, hazard ratio.

(Table 3), we carried out analyses excluding individuals prescribed psychotropic (i.e., antidepressants or benzodiazepines) or cardiac medications (i.e., calcium channel blockers, renin-angiotensin system acting agents, or statins), and individuals with polypharmacy (i.e., 5 or more different medication classes during the same calendar year). Associations remained similar to the main analyses when excluding individuals with each respective medication or polypharmacy.

To examine if associations varied by cardiac severity, we analysed those who had been hospitalised for cardiac conditions within 1 year of medication starting and all others separately. Hospitalised individuals showed increased hazards of psychiatric hospitalisations (HR: 1.14, 95% CI: 1.12 to 1.17, $p < 0.001$) and suicidal behaviour (HR: 1.20, 95% CI: 1.05 to 1.36, $p = 0.006$), and a decreased risk of violent crime (HR: 0.85, 95% CI: 0.73 to 0.98, $p = 0.022$) during β-blocker treatment. Non-hospitalised individuals demonstrated decreased hazards of psychiatric hospitalisations and violent crime, and no associations with suicidal behaviour.

We also examined individuals with and without a history of psychiatric disorders and/or suicidal behaviour before the start of the study period separately. Results remained similar to the main results for those with a history of psychiatric disorders and/or suicidal behaviour. For those without, hazards were decreased for psychiatric hospitalisations and violent crime and did not reach statistical significance for suicidal behaviour (HR: 1.03, 95% CI: 0.93 to 1.15, $p = 0.566$) during β-blocker treatment.

Finally, we carried out sensitivity analyses to further examine the robustness of associations with violent crime (Table 3). First, we examined only those with a history of violent crime before the start of the study period to assess if β-blocker treatment periods were differentially associated with violence in this group. We found reduced hazards of violent crime (HR: 0.82, 95% CI: 0.74 to 0.91, $p < 0.001$) during β-blocker treatment. Second, we stratified associations by different age groups, up to age 30, 30 to 49, 50 to 60, and 70 and older. We found reduced hazards of violent crime for all age groups during β-blocker treatment periods, although hazards did not reach statistical significance for the 2 younger groups (HR: 0.78, 95% CI: 0.59 to 1.03, $p = 0.077$; HR: 0.76, 95% CI: 0.56 to 1.03, $p = 0.073$).

## Discussion

In this population-based cohort of 1.4 million persons in Sweden who had been treated with β-blockers between 2006 and 2013, we used a within-individual design that accounted for background factors associated with confounding by indication. We found some heterogeneity in the direction of associations of β-blockers with the psychiatric and behavioural outcomes investigated; notably, we found that periods on β-blocker treatment were associated with decreased psychiatric hospitalisation hazards (HR: 0.92, 95% CI: 0.91 to 0.93, $p < 0.001$) as compared to periods off treatment. In addition, there was a 13% (HR: 0.87, 95% CI: 0.81 to 0.93, $p < 0.001$) lower risk of being charged with a violent crime by the police or prosecution services during β-blocker treatment. In contrast, there was a small increased association with treatment for suicidal behaviour and suicide mortality (HR: 1.08, 95% CI: 1.02 to 1.15, $p = 0.012$; with 0.7% of the cohort experiencing this outcome during the study period) during β-blocker treatment. We carried out several sensitivity analyses to test the robustness of results, and reduced associations with violent crime during β-blocker treatment periods were consistent. However, links with reduced psychiatric hospitalisations and increased suicidal behaviour during β-blocker treatment shown in the principal analyses were not consistent across all sensitivity analyses, suggesting that these findings could be partially confounded.

Prior studies on β-blockers and psychiatric outcomes have failed to adjust for the effect of co-medications [5]. In our study, associations with psychiatric hospitalisations and violent crime remained when excluding individuals prescribed other anti-hypertensive or psychotropic medications. Furthermore, the majority of observational studies have included prevalent β-blocker users. However, if the risk of outcomes varies with time after treatment initiation, including prevalent users could introduce bias [70]. When we excluded prevalent users from our analyses, associations remained similar for all outcomes.

The mechanism of action of β-blockers on aggression is uncertain; possible explanations include mild sedation [71], or reduced adrenergic activity at the central or peripheral level, resulting in decreased catecholaminergic reactions (i.e., "fight or flight") to stressful situations [40–42]. We found the reduced associations with violent crime charges during β-blocker treatment were consistent using alternative time periods, excluding individuals with co-prescribed medications, excluding prevalent users, stratifying by different age groups, and stratifying on hospitalisations for cardiac conditions. The latter would address the potential explanation that individuals with severe cardiac conditions might be more incapacitated, and therefore less likely to commit a violent crime. However, we found that associations remained decreased in both those hospitalised and those not. Our results were broadly consistent with evidence from small studies on individuals with psychiatric conditions and cognitive impairment [40,41,72], but we have substantially increased the sample size. We also showed reductions for non-violent crime during β-blocker treatment and for violent crime in 2 higher-risk groups, i.e., those with a history of psychiatric problems and violent crime, respectively. As evidence-based

treatments for violent outcomes are very limited, this is a potentially important finding [73]. Currently, individuals are prescribed β-blockers for aggression in psychiatric clinics and hospitals, and the current work suggests some support for this. This is underscored by absolute rates of violent crime charges—in those with a psychiatric history in the β-blocker cohort (*n* = 92,619), there were 7,502 violent crime charges during the study period committed by 2.3% (*n* = 2,153) of this subcohort. Importantly, the current work is consistent with 2 small RCTs of β-blockers (propranolol and nadolol) on violent outcomes in psychiatric patients [74].

We found reduced associations with psychiatric hospitalisations during β-blocker treatment periods. Importantly, we included a wider range of psychiatric disorders than in previous studies, and our results suggest that β-blockers are associated with reductions in severe psychiatric disorders (i.e., that lead to a hospitalisation). We carried out separate analyses for 3 groups of psychiatric disorders previously linked to β-blocker use: psychotic, depressive, and anxiety disorders. Our results showed decreased hazards of psychotic disorder hospitalisations during periods of β-blocker treatment, in contrast to previous case reports proposing an increased risk [32]. We also found reduced hazards of hospitalisations for depressive disorders during β-blocker treatment, which is consistent with other observational studies [16–18,75]. However, we found no associations between β-blocker treatment and hospitalisations for anxiety disorders, although they are widely prescribed for this. This is in line with other work showing that β-blockers lead to little improvement in the long-term treatment of anxiety disorders [2,29,76]. However, the findings on depression and anxiety are complicated because these registers will selectively include more severe cases of these disorders, which come to the attention of secondary services. Thus, it is possible that there is a reduction in anxiety not identified in this study. At the very least, our findings suggest no association with severe cases of anxiety. As for depression, the findings, if triangulated, may suggest some benefits in severe depression. β-blockers have been proposed to have antidepressant properties, either by reducing inflammation or by binding to serotonin receptors [17,68]; however, the precise mechanism of action is unknown. The results on psychiatric outcomes need consideration of absolute risks—overall, psychiatric hospitalisations were 6.9% in the cohort, which is not small, but for individual disorders, absolute hospitalisation rates were low (0.6% for psychosis, 2.4% for depression, and 1.9% for anxiety).

We found a small increased link with suicidal outcomes during β-blocker treatment periods, consistent with previous observational studies [11–13]. However, this was specific to individuals with a history of psychiatric hospitalisations or suicidal behaviour, and the absolute risk was low (at 0.7%). One explanation is that individuals with past psychiatric problems may be at risk of a suicidal outcome when they experience a cardiac condition (and consequently, are treated with β-blockers). Several psychological reactions are reported to occur after a cardiac event that can affect mood [77]; individuals may have negative thoughts about their overall well-being, be uncertain about the future, concerned about reduced physical ability, or feel guilty about previous habits that may have increased the risk of the cardiac event. In line with this, research shows that the risk of suicide is increased during the first months after a cardiac event [48,78], and one explanation for our findings could be that the psychological burden associated with the cardiac condition, rather than the β-blocker treatment, increases suicidal risk. The risk of suicidal behaviour also remained increased when we excluded the first 3 months of incident β-blocker treatment, which would suggest a prolonged risk period, as proposed in previous research [48,78].

However, the findings on psychiatric hospitalisations and suicidal behaviour were not consistent in some sensitivity analyses. The main difference was increased hazards for both psychiatric hospitalisations and suicidal behaviour among those hospitalised for cardiac conditions

(14% and 20%, respectively). Severe heart failure has been linked to an increased risk of depression and suicide [46,79], and these results suggest that severe cardiac problems, rather than the β-blocker treatment, increase the risk of serious psychiatric events.

We also examined if associations could be attributed to nonspecific treatment effects, such as increased supervision or healthcare contacts, by using another cardiac medication (ACE inhibitors) and a non-cardiac medication (antihistamines) as independent exposures in the β-blockers cohort. In these analyses, we found no clear associations with psychiatric hospitalisations or violent crime, and small increased links with suicidal behaviour. If associations were to be confounded by nonspecific treatment effects, we would have expected similar patterns for all outcomes during treatment with ACE inhibitors and antihistamines, as during β-blocker treatment. The differing treatment patterns for psychiatric hospitalisations and violent crime suggest that nonspecific treatment effects were not prominent. The increased links with suicidal behaviour could suggest that associations were not specific (i.e., causally related) to β-blockers.

Moreover, we stratified analyses by selectivity, solubility, and by individual β-blockers, and found some differences. However, due to multiple testing, we interpret our findings with caution. One possibility is that hydrophilic β-blockers (such as atenolol) are more favourable for treating psychiatric outcomes, which has previously been proposed [3,5,25].

Strengths include a large, population-based cohort of 1.4 million individuals treated with β-blockers over 8 years that is representative of β-blocker users, using outcomes from validated, high-quality registers with nationwide coverage, and having complete information on β-blocker dispenses, as each prescription collected at the pharmacy was registered. We used a within-individual design that controls for stable covariates, such as genetics or early background factors, and carried out several sensitivity analyses, including the use of 2 negative control medications as independent exposures to examine nonspecific treatment effects. Important limitations include that this was an observational study, and caution needs to be exercised when drawing causal inferences. Even though our model adjusted for stable factors associated with confounding by indication to a larger extent than models that compare users to non-users, it did not account for confounders that could change during treatment (such as nonspecific treatment effects), unless measured and adjusted for in the model. The use of official registers involves selection effects and will underestimate rates of underlying disorders and outcomes. Using secondary care and mortality outcomes will selectively include more severe cases of disorders, thus our results may not generalise to less severe cases and/or cases that were not diagnosed by specialists in psychiatry. On the other hand, official registers capture information on actual healthcare contacts, reflecting real-world outcomes that consume resources. Differences between countries might affect the generalisability of findings; in 2019, Sweden had 1,708 in-patient hospital discharge rates for circulatory diseases per 100,000 inhabitants (range for EU member states: 930 per 100,000 to 4,697 per 100,000) [80]. Deaths due to diseases of the circulatory system constituted 32.1% of all deaths in 2018 (range for EU member states: 21.6% to 65.4%) [80]. In a study of primary care practices in 14 European countries, 32% of patients with chronic heart failure in Sweden were prescribed β-blockers (mean in the 14 countries: 20%) [81]. Although data on β-blockers was based on individuals collecting their medication from pharmacies, which is an advance from prescription-only data, medication adherence was not known. To address this, we only included individuals with at least 2 collected prescriptions within 6 months, and we also excluded individuals who were instructed to take the medications as required. Furthermore, the Prescribed Drug Register started in July 2005. We carried out sensitivity analyses excluding prevalent users (by examining only those who initiated β-blocker treatment from January 1, 2007 and onwards); however, these individuals could have been treated before the start of the Prescribed Drug Register. Nevertheless, our analyses included a β-blocker washout period of 18 months. In our

primary analyses, we defined the end of a treatment period as the day of the last dispensed prescription, which gives a more conservative estimate of medication exposure. However, our sensitivity analyses accounting for discontinuation or late treatment effects showed no differences in associations. Finally, differences between countries in prescription patterns, including indications for the prescriptions, might affect the generalisability of findings.

Our findings demonstrated reduced associations with charges for violent crimes during β-blocker treatment. More studies using other designs (e.g., randomised controlled trials) are needed to better understand the role of β-blockers in the management of aggression and violence. In addition, the use of β-blockers to manage anxiety is not supported in this real-world study of new presentations of anxiety in secondary patient care. If triangulated using other designs, β-blockers could be used to manage aggression and hostility in individuals with psychiatric conditions.

## Supporting information

**S1 Checklist. STROBE Statement.**
(PDF)

**S1 Fig. Flowchart of cohort inclusion.**
(TIFF)

**S2 Fig. Age-adjusted within-individual associations using stratified Cox proportional hazards regression between β-blockers and psychiatric and behavioural outcomes stratified by individual β-blockers.**
(TIFF)

**S1 Table. Treatment characteristics during the study period (2006–2013).**
(TIFF)

**S2 Table. Number of events stratified by selectivity, solubility, indication for the prescription, and by individual β-blockers.**
(TIFF)

**S3 Table. Symptoms and disorders included in each indication[†].**
(TIFF)

**S4 Table. Unadjusted within-individual associations using stratified Cox proportional hazards regression between β-blockers and psychiatric and behavioural outcomes.**
(TIFF)

**S1 Text. Supplementary material to the manuscript.**
(DOCX)

## Acknowledgments

We thank Remus-Giulio Anghel for assistance with annotation and programming of indications.

## Author Contributions

**Conceptualization:** Yasmina Molero, Sam Kaddoura, Ralf Kuja-Halkola, Henrik Larsson, Paul Lichtenstein, Brian M. D'Onofrio, Seena Fazel.

**Data curation:** Yasmina Molero.

**Formal analysis:** Yasmina Molero.

**Funding acquisition:** Henrik Larsson, Paul Lichtenstein, Brian M. D'Onofrio, Seena Fazel.

**Methodology:** Yasmina Molero, Ralf Kuja-Halkola, Seena Fazel.

**Project administration:** Yasmina Molero.

**Resources:** Henrik Larsson, Paul Lichtenstein, Brian M. D'Onofrio.

**Supervision:** Seena Fazel.

**Visualization:** Yasmina Molero.

**Writing – original draft:** Yasmina Molero, Seena Fazel.

**Writing – review & editing:** Yasmina Molero, Sam Kaddoura, Ralf Kuja-Halkola, Henrik Larsson, Paul Lichtenstein, Brian M. D'Onofrio, Seena Fazel.

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
