## [Editor Report · Decision Letter 0]

6 Jul 2022

Dear Dr Fazel, 

Thank you for submitting your manuscript entitled "Associations between β-blockers and psychiatric and behavioural outcomes – a population-based study of 1.4 million individuals" for consideration by PLOS Medicine.

Your manuscript has now been evaluated by the PLOS Medicine editorial staff and I am writing to let you know that we would like to send your submission out for external peer review.

Please re-submit your manuscript within two working days, i.e. by Jul 08 2022 11:59PM.

Kind regards,

Caitlin Moyer, Ph.D.

Associate Editor

PLOS Medicine

---

## [Decision Letter · Decision Letter 1]

12 Oct 2022

Dear Dr. Fazel,

Thank you very much for submitting your manuscript "Associations between β-blockers and psychiatric and behavioural outcomes – a population-based study of 1.4 million individuals" (PMEDICINE-D-22-02228R1) for consideration at PLOS Medicine. 

[LINK]

In light of these reviews, I am afraid that we will not be able to accept the manuscript for publication in the journal in its current form, but we would like to consider a revised version that addresses the reviewers' and editors' comments. Obviously we cannot make any decision about publication until we have seen the revised manuscript and your response, and we plan to seek re-review by one or more of the reviewers. 

We expect to receive your revised manuscript by Nov 01 2022 11:59PM. Please email us (plosmedicine@plos.org) if you have any questions or concerns.

We look forward to receiving your revised manuscript. 

Sincerely,

Philippa Dodd, MBBS MRCP PhD

PLOS Medicine

plosmedicine.org

GENERAL

Please address all reviewer and editor comments detailed below

Please remove the Header from each page

Please ensure that the study is reported according to the STROBE guideline, and include the completed STROBE checklist as Supporting Information. Please add the following statement, or similar, to the Methods: "This study is reported as per the Strengthening the Reporting of Observational Studies in Epidemiology (STROBE) guideline (S1 Checklist)." The STROBE guideline can be found here: http://www.equator-network.org/reporting-guidelines/strobe/

Please remove the data availability statement and competing interests statement from the end of the manuscript and include only in the submission form

TITLE

Please revise your title according to PLOS Medicine's style. Your title must be nondeclarative and not a question. It should begin with main concept if possible. "Effect of" should be used only if causality can be inferred, i.e., for an RCT. Please place the study design ("A randomized controlled trial," "A retrospective study," "A modelling study," etc.) in the subtitle (ie, after a colon).

ABSTRACT

* Please structure your abstract using the PLOS Medicine headings (Background, Methods and Findings, Conclusions). 

* Please combine the Methods and Findings sections into one section, “Methods and findings”.

Abstract Methods and Findings: 

* Please ensure that all numbers presented in the abstract are present and identical to numbers presented in the main manuscript text. 

* Please include the study design, population and setting, number of participants, years during which the study took place, length of follow up, and main outcome measures. 

* Please quantify the main results p-values as well as with 95% CIs. To improve reader accessibility I would suggest removal of the “=” symbol and present your data as follows: “(HR: 0.87, 95% CI: 0.81- 65 0.93, p<0.01)” for example

* Please include the important dependent variables that are adjusted for in the analyses. 

* In the last sentence of the Abstract Methods and Findings section, please describe the main limitation(s) of the study's methodology (as opposed to the limitations of the observational nature of the study).

AUTHOR SUMMARY

INTRODUCTION

Please remove the sub-heading “Aims” line 138 and conclude the introduction with a clear description of the study question/hypotheses.

METHODS and RESULTS

In the manuscript text, please ensure you have included the following

(1) the specific hypotheses you intended to test, 

(2) the analytical methods by which you planned to test them, 

(3) the analyses you actually performed, and 

(4) when reported analyses differ from those that were planned, transparent explanations for differences that affect the reliability of the study's results. If a reported analysis was performed based on an interesting but unanticipated pattern in the data, please be clear that the analysis was data-driven.

Did your study have a prospective protocol or analysis plan? Please state this (either way) early in the Methods section. 

As in the abstract, where statistical data are reported, for example line 350 “(HR=1.08, 95% CI=1.02-1.15)” please include p-values

please report data as described previously with the absence of “=” symbol i.e. “(HR: 1.08, 95% CI: 1.02- 1.15, p<0.01)”

where p-values are reported please also include the statistical test used to determine them

TABLES

Please re-title table 1 to read “Baseline characteristics of the study cohort” or something similar

Where 95% CIs reported please also include p-values. In the table caption/legend please state the statistical test used to determine them 

FIGURES

For all figures please include an appropriate figure caption/legend which appropriately describes the data presented in the figures. Please include and define any abbreviations. Please check and amend throughout all figures (and tables) including those in the supplementary files.

DISCUSSION

Please remove the headings “strengths and limitations” and conclusions from below the discussion and structure the discussion as follows: a short, clear summary of the article's findings; what the study adds to existing research and where and why the results may differ from previous research; strengths and limitations of the study; implications and next steps for research, clinical practice, and/or public policy; one-paragraph conclusion.

REFERENCES

Please ensure you have followed our guidelines for listing references which can be found here: https://journals.plos.org/plosmedicine/s/submission-guidelines#loc-references

Journal name abbreviations should be those found in the National Center for Biotechnology Information (NCBI) databases. 

SOCIAL MEDIA

In the event that your manuscript is published, to help us extend the reach of your research, please provide any Twitter handle(s) that would be appropriate to tag, including your own, your coauthors’, your institution, funder, or lab. Please respond to this email with any handles you wish to be included when we tweet this paper.

Comments from the Academic Editor:

Its a really interesting paper and addresses an important (if untrendy) topic with an amazing database. The findings have potential to influence future research and practice. They have been careful with their interpretation and rigorous with sensitivity analyses. As long as the statistical analytical concerns are not fatal, it looks as if the reviewer's points can all be addressed. The reviewers were thorough and have articulated most of the issues that I noted. I had one confusion which you might ask the authors to address (unless you can see that they address it and I missed it):

- how were repeated events of the same type of outcome in one individual handled within the analysis? 

Comments from the reviewers:

Reviewer #1: This is an interesting population-based study on the associations between β-blockers and psychiatric and behavioural outcomes. However, there are a few major issues needing attention.

1) Authors said "We applied within-individual Cox proportional hazards regression to compare periods on treatment with periods off treatment in order to reduce possible confounding by indication". Using self controls is an alternative to standard epidemiology method, but not sure and not convinced how to apply this within the Cox model framework. There is no methodogical reference on this within-individual cox model especially statistically. As it's a time to event analysis, if a patient have multiple on and off treatment, how can you define the time? added up? Any washout time between on and off treatment? Any long term effect of Beta blocker after off treatment? But that becomes unnatural and interrupted. Also how about proportional hazard assumption? Censoring? Cox model was not designed for this type of self control and I would like to see the theoretical proof and justification of applying Cox model in this setting properly.

2) Competing risk. For outcomes other than all cause mortality in survival analyses, competing risk (from death) need to be considered and adjusted using methods such as fine and grey model. However, this issue was not considered at all in the paper, therefore the resulted HRs could be inaccurate and need to be adjusted for competing risk.

3) Quite a few writing in the paper is difficult to follow and confusing, such as in the findings in the abstract: "There was heterogeneity in the direction of results; there were reductions in psychiatric hospitalisations...". Then, what compares what? Normally we would say "treatment A, comparing to treatment B, reduced the risk of hospitalisation by x%". There are many places in the paper where the interpretation of HRs needs to improve.

Reviewer #2: This is an interesting longitudinal study using linked registry data that examines the associations between beta blockers and psychiatric events in Sweden. The study and it's findings suggest that beta blockers contribute to or are associated with increases in suicidal risks but not anxiety or violent crimes.

I have several comments that warrant some revisions to strengthen the paper.

1. It is not clear why authors report psychiatric outcomes in aggregate. It is clear from prior literature that beta blockers are associated with suicidal symptom or depression but less on anxiety and criminal

behavior. This study reinforces this given heterogeneity in effects. 

2. The rationale for the use of antihistamines (vs other meds)as a control is not clear and potentially problematic l. First antihistamines is a therepautic category with many very different drugs. Second, antihistamines—many of them- are available over the counter and not clear if the medication data captures over the counter dispensings or sales. Third, why not use an antihypertensive and limit to older age groups? For example ace inhibitors?

3. Not clear how analyses accounted for the initiation or dispensing of antidepressants. This needs to clarified and addressed.

Reviewer #3: Thank you for the opportunity to review this important paper. I did not review the original version, so my comments are from a 'first read' perspective. You have carefully presented a series of complex and well-thought out SCCS, and epidemiologists (and hopefully clinicians!) will enjoy reading this from both the clinical-implications and methodological perspectives. You have answered most of the questions I had as the manuscript went on, and present a balanced conclusion taking all analyses into account. 

Comments for your consideration:

-The introduction provides a sound overview of the literature. The only missing comment might be about adverse drug reaction labelling of b-blockers-as some psychiatric events and sleep disorders are listed in SPC as potential ADRs. 

-I understand that you have included information about registries in the supplementary material, but a brief comment about linked data sources in the design, will also be helpful. 

-A stronger justification as to why two dispenses were required for inclusion in the cohort will be beneficial. There is potential for survival bias if individuals stopped taking beta-blocker after first use, due to adverse events, and the exclusion of these individuals might direct associations towards the null.

-Ending exposure date on prescription date will be an underestimate of exposure, and may introduce potential for misclassification. I am not sure whether all readers would agree with this choice of definition, but you do address this and provide assurance via sensitivity analyses.

-Further justification of excluding prn use is required given as this is likely to exclude people prescribed propranolol for prn use for anxiety.

-My understanding is that there is no primary care diagnostic data, unless this is captured on prescription direction in Sweden (this rarely happens in the UK). How confident can you be of diagnostic classification? A comment about the lack of primary care exposure diagnoses and outcome data (only severe outcomes reported-hospitalisation and death) would be beneficial, including the direction of effect on estimates. 

-Ethical approval is 9 years old-I presume that you have pan-database approval and this is why the approvals are old. If this is the case, please state this along with how, and when, this individual study was approved.

-Despite your justification in S1, I am unsure antihistamines were the most appropriate choice of negative control, due to seasonal and ad hoc use, and possible misclassification to non-prescription self medication. That said, you have included plenty of thoughtful sensitivity analyses.

-A few times, you discuss the potential involvement of suicidality following major cardiac events. Perhaps a comment about the psychological burden of sudden and major, life-changing health conditions is warranted, which might be an important factor accounting for any association. 

Reviewer #4: PMEDICINE-D-22-02228R1 

In this study 1,400,766 individuals aged 15 years or older who had collected β-blocker prescriptions were included and followed them between 2006 and 2013 in healthcare, mortality, and crime registers. After exclusions due to irregular medication use and age (S1 311 Fig), the final cohort included 1,400,766 individuals (15.7% of the total population of Sweden aged 15 years or older during the study period [n=8,945,456]). During the study 329 period, 6.9% (n=96,801) of the β-blocker users were hospitalised for a psychiatric disorder, 330 0.7% (n=9,960) presented with suicidal behaviour (i.e. self-injurious acts, suicide attempts, β-blockers and psychiatric and behavioural outcomes and deaths from suicide), and 0.7% (n=9,405) were charged with a violent crime.

Register data on dispensed βblocker prescriptions were linked with with hospitalisations for psychiatric disorders, suicidal behaviour (including deaths from suicide), and charges of violent crime using nationwide registers. Within-individual Cox proportional hazards regressionwere performed to compare periods on treatment with periods off treatment in order to reduce possible confounding by indication. In further analyses, adjustments by stated indications, prevalent users, cardiac severity, psychiatric history, individual β-blockers, β-blocker selectivity and solubility, and other medications were made.

In this population-wide study, we found no consistent links between β-blockers and psychiatric outcomes. However,iincreased associations were found with suicidal behaviour 350 during β-blocker treatment periods (HR=1.08, 95% CI=1.02-1.15), and reduced associations 351 with violent crime (HR=0.87, 95% CI=0.81-0.93).Thus it was concluded that β-blockers were associated with reductions in violence, which remained in sensitivity analyses. The use of β-blockers to manage aggression and violence could be investigated further.

As mentioned, limitations include that this is an observational study, and caution needs to be exercised when drawing causal inferences.

Reviewer's comments

Manuscript PMEDICINE-D-22-02228R1 describes an very nice observational study using large cohorts from Swedish national registries investigating β-blockers and psychiatric disorders, suicidal behaviour and violent crime.. Although highly interesting some questions arise:

Comment 1: The following registers were consulted: the Total Population Register, the Longitudinal integrated database for health insurance and 214 labour market studies (LISA), the National patient register, the register of people Suspected of Offences and the Swedish prescribed drug register. 

These are not described in the text, and also not in the Supplement text as stated. It is strongly advised to briefly describe these registers in a few sentences in the text.

Comment 2: How were 'suicidal behaviour' and 'violent crime assessed'? Only as yes/no (it seems) or with questionnaires? The results should be presented in the manuscript. Now it is only represented as number of events, but this should be explained in depth.

Comment 3. The association of b-blocker treatment and reduction in violent crime puzzles me. The total patient group consisted of mostly older people, (>40% over 70 years) and lots of retired people. It could be speculated that the people with crime pasts were less violent because they aged. Although a sensitivity analysis of this special subgroup was performed, it is recommended to also look into this and describe this. Another point is that although some other co-medications were assessed, also benzodiazepines, which could result is less violence, should be assessed. Both of these should be addressed in in the manuscript.

Comment 4: When investigating drug effects, it is always important to assess co-medication and especially polypharmacy, as interactions may occur that influence behavior. Was this assessed? It is shortly mentioned but given it's potential importance as a confounder results should be shown and it should be described in more detail in the manuscript.

[LINK]

---

## [Decision Letter · Decision Letter 2]

13 Dec 2022

Dear Dr. Fazel,

Thank you very much for re-submitting your manuscript "Associations between β-blockers and psychiatric and behavioural outcomes: A population-based cohort study of 1.4 million individuals in Sweden" (PMEDICINE-D-22-02228R2) for review by PLOS Medicine.

I have discussed the paper with my colleagues and the academic editor and it was also seen again by 3 reviewers. I am pleased to say that provided the remaining editorial and production issues are dealt with we are planning to accept the paper for publication in the journal.

[LINK]

We look forward to receiving the revised manuscript by Dec 20 2022 11:59PM.   

Sincerely,

Philippa Dodd, MBBS MRCP PhD

PLOS Medicine

plosmedicine.org

Requests from Editors:

GENERAL

Thank you for your exceptionally detailed and considerate response to editor and reviewer requests. Please see below for further minor changes which we request that you address in full.

We note the reviewer comment regarding language and clarity and have some suggestions below, otherwise our copyediting team will help with minor grammatical revisions, as necessary.

REQUEST TO CHANGE AUTHORSHIP

We accept the request to add the author to the author list given the described contribution made. We thank you for including the letter signed by all co-authors which demonstrates that they consent to the change in authorship. 

ABSTRACT

Thank you for including p-values and revising the presentation of your statistical reporting please report p as <0.001 or 0.012, as opposed to <.001. Please check and mend throughout all sections of the manuscript.

AUTHOR SUMMARY

Line 95: “treated with β-blockers using a within-individual design…” it may be apparent to the general reader what a within individual design refers to. Would perhaps benefit from brief clarification or use of a more “lay” term (you do this very nicely in the introduction at lines 178-179)

Similarly, line 108: “If findings on violence are triangulated using other designs…” what does it mean to triangulate findings?

METHODS and RESULTS

Line 194: 2…on the registers, see S1 Text, p. 2.” When referring to the supplementary data please use “page” instead of “p.” – please check and amend throughout.

Line 463 onwards: please report p-values as <0.001 not .001

TABLES and FIGURES

Throughout, in all tables and, as above, please report p as <0.001

Table 1: Under the characteristic “sex” this should read female and male as opposed to women and men

For the purposes of transparent data reporting, PLOS Medicine requests that where adjusted analyses are presented, unadjusted analyses are reported concurrently. 

For figures 1, 2, 3 & S2 and tables 2 & 3 please indicate whether your analyses are adjusted and if so, please detail in the table/figure caption or footnote which factors are adjusted for. 

Please also include the unadjusted analyses for comparison. 

Please ensure that p-values are reported (presented as described above) alongside 95% CIs. 

Figure 1,2,3 & S2: please indicate in the figure caption/footnote what the dots and lines represent

REFERENCES

Throughout, please ensure that citations are placed in square brackets

Line 349: “…phase following a cardiac event (69, 70)…” please remove spaces between citations as follows: [69,70]

Please check and amend throughout

SOCIAL MEDIA

Please include your twitter handles in the manuscript submission form (if not already done so)

Comments from Reviewers:

Reviewer #1: Thanks authors for their great effort to improve the manuscript. All my comments were well addressed. I am satisfied with the response and revision. No further issues needing attention.

Reviewer #2: The authors adequately addressed reviewers comments but manuscript may benefit from some editing for language and clarity. 

Reviewer #4: As stated before Manuscript PMEDICINE-D-22-02228R1 describes an very nice observational study using large cohorts from Swedish national registries investigating β-blockers and psychiatric disorders, suicidal behaviour and violent crime.

The authors took all the suggestions earlier made into account and not only added the suggested additional information, but also performed additional sensitivity analysis.

It is a pleasure to review when authors see the added value of adding information and analyses even though it takes extra work. 

The proposed adjusted manuscript seems improved and acceptable for publication in it's current form.

[LINK]

---

## [Editor Report · Decision Letter 3]

28 Dec 2022

Dear Dr Fazel, 

On behalf of my colleagues and the Academic Editor, Professor Charlotte Hanlon, I am pleased to inform you that we have agreed to publish your manuscript "Associations between β-blockers and psychiatric and behavioural outcomes: A population-based cohort study of 1.4 million individuals in Sweden" (PMEDICINE-D-22-02228R3) in PLOS Medicine.

Prior to publication please ensure that the following final revision has been made:

* In an appropriate part of the main manuscript text, please signpost the reader to the unadjusted analyses in table S4, we thank you for including them. Suggest perhaps the end of line 475 (1st paragraph of your results section) following reporting of the baseline characteristics of the study cohort.

PRESS

Best wishes, 

Philippa Dodd, MBBS MRCP PhD 

PLOS Medicine